# Plasma Gelsolin Reinforces the Diagnostic Value of FGF-21 and GDF-15 for Mitochondrial Disorders

**DOI:** 10.3390/ijms22126396

**Published:** 2021-06-15

**Authors:** Ana Peñas, Miguel Fernández-De la Torre, Sara Laine-Menéndez, David Lora, María Illescas, Alberto García-Bartolomé, Montserrat Morales-Conejo, Joaquín Arenas, Miguel A. Martín, María Morán, Cristina Domínguez-González, Cristina Ugalde

**Affiliations:** 1Mitochondrial and Neuromuscular Disorders Research Group, Hospital 12 de Octubre Research Institute (imas12), 28041 Madrid, Spain; anappdv@gmail.com (A.P.); miguel.fnandezt.imas12@h12o.es (M.F.-D.l.T.); slaine.imas12@h12o.es (S.L.-M.); maria.illescas@hotmail.com (M.I.); alberto7188@hotmail.com (A.G.-B.); montserrat.morales@salud.madrid.org (M.M.-C.); joaquin.arenas@salud.madrid.org (J.A.); mamcasanueva.imas12@h12o.es (M.A.M.); mmoran@h12o.es (M.M.); 2Clinical Research Unit, Hospital 12 de Octubre Research Institute (imas12), 28041 Madrid, Spain; david@h12o.es; 3Departamento de Estadística y Ciencia de los Datos, Facultad de Estudios Estadísticos, Universidad Complutense de Madrid, 28040 Madrid, Spain; 4Centro de Investigación Biomédica en Red de Epidemiología y Salud Pública (CIBERESP), 28029 Madrid, Spain; 5Congenital Metabolic and Minority Diseases Unit, Internal Medicine Service, Hospital 12 de Octubre, 28041 Madrid, Spain; 6Centro de Investigación Biomédica en Red de Enfermedades Raras (CIBERER), 28029 Madrid, Spain; 7Department of Clinical Biochemistry, Hospital 12 de Octubre, 28041 Madrid, Spain; 8Neuromuscular Unit, Department of Neurology, Hospital 12 de Octubre, 28041 Madrid, Spain

**Keywords:** mitochondrial disorders, OXPHOS deficiency, biomarkers, plasma GSN, FGF-21, GDF-15

## Abstract

Mitochondrial disorders (MD) comprise a group of heterogeneous clinical disorders for which non-invasive diagnosis remains a challenge. Two protein biomarkers have so far emerged for MD detection, FGF-21 and GDF-15, but the identification of additional biomarkers capable of improving their diagnostic accuracy is highly relevant. Previous studies identified Gelsolin as a regulator of cell survival adaptations triggered by mitochondrial defects. Gelsolin presents a circulating plasma isoform (pGSN), whose altered levels could be a hallmark of mitochondrial dysfunction. Therefore, we investigated the diagnostic performance of pGSN for MD relative to FGF-21 and GDF-15. Using ELISA assays, we quantified plasma levels of pGSN, FGF-21, and GDF-15 in three age- and gender-matched adult cohorts: 60 genetically diagnosed MD patients, 56 healthy donors, and 41 patients with unrelated neuromuscular pathologies (non-MD). Clinical variables and biomarkers’ plasma levels were compared between groups. Discrimination ability was calculated using the area under the ROC curve (AUC). Optimal cut-offs and the following diagnostic parameters were determined: sensitivity, specificity, positive and negative predictive values, positive and negative likelihood ratios, and efficiency. Comprehensive statistical analyses revealed significant discrimination ability for the three biomarkers to classify between MD and healthy individuals, with the best diagnostic performance for the GDF-15/pGSN combination. pGSN and GDF-15 preferentially discriminated between MD and non-MD patients under 50 years, whereas FGF-21 best classified older subjects. Conclusion: pGSN improves the diagnosis accuracy for MD provided by FGF-21 and GDF-15.

## 1. Introduction

The mitochondrial oxidative phosphorylation (OXPHOS) system provides most of the chemical energy (ATP) usable by cells and tissues through the concerted action of five multiprotein enzyme complexes (CI to CV) and two mobile electron carriers (ubiquinone and cytochrome *c*) [1]. Functional defects of the OXPHOS machinery are the main cause of inborn errors of energy metabolism, comprising more than 250 disorders with high genetic and clinical variability, globally affecting 1:5000 newborns [2]. The clinical symptoms include highly heterogeneous phenotypes encompassing dysfunction of almost any organ system, which can occur at any age [3]. In recent years, next-generation sequencing (NGS) has focused most diagnostic efforts on the discovery of new disease-causing genes [4], but the pathophysiological mechanisms underlying mitochondrial OXPHOS disorders (MD) remain mostly unknown. As a consequence, specific and sensitive diagnostic tools for MD based on serum biomarkers are scarce. Only the cytokines fibroblast-growth factor 21 (FGF-21) and growth-differentiation factor 15 (GDF-15) are so far considered as promising indicators of mitochondrial disorders, especially those with muscle involvement [5,6,7]. FGF-21 is a growth factor with regulatory roles in glucose and lipid metabolism that is produced mostly by the liver [8,9]. It was described as a serum biomarker for muscle-manifesting MD, where it also positively correlates with lactate levels and disease severity [10,11], and for patients carrying either mutations in nuclear genes involved in mitochondrial DNA (mtDNA) maintenance [12] or single mtDNA deletions [13]. However, FGF-21 does not discriminate non-myopathic MD, such as LHON or sensorineural deafness [14], or disease progression in patients harboring the MELAS-associated m.3243A > G mutation [15,16]. High FGF-21 levels have been also linked to many other metabolic pathologies, such as diabetes mellitus, obesity, or chronic kidney disease [17,18,19]. Alternatively, GDF-15 is a member of the transforming growth factor beta superfamily, which is highly expressed in placenta [20]. Increased serum GDF-15 levels have been associated with MD originated by different mutations and clinical conditions [21,22,23,24]. In different cohorts of MD patients, GDF-15 positively correlated with FGF-21, but usually showing a higher sensitivity for MD diagnosis than the latter [23,25,26,27]. Notably, GDF-15 levels also increase as a response to stress and inflammation, and as a result, the blood concentration of GDF-15 is elevated in a variety of cardiovascular diseases, including heart failure and atherosclerosis, as well as in insulin resistance, renal insufficiency [28], and multiple sclerosis [29]. The combination of FGF-21 and GDF-15 was proposed to increase efficiency for identifying MD patients more than either factor alone [26]. However, their specificity for MD diagnosis was challenged by studies showing elevated levels of both markers in a variety of inherited metabolic diseases of non-mitochondrial origin, especially those with liver involvement [30]. Since none of these biomarkers are absolutely specific for MD, the identification of additional blood biomarkers that improve their diagnostic value is a challenging task in the field [31].

Previous proteomics-based studies revealed Gelsolin (GSN), a cytoskeletal protein that regulates the maintenance and dynamics of actin filaments [32], as a potential therapeutic target for OXPHOS dysfunction [33]. The human *GSN* gene is located at chromosome 9 [34], and its alternative transcription initiation and mRNA processing lead to the translation of two main protein isoforms: the cytoplasmic (cGSN) and plasma (pGSN) isoforms (UniProtKB references P06396-1 and P06396-2, respectively) [35,36]. pGSN is mainly secreted from muscle into the bloodstream [37]. It differs from cGSN by the presence of a 51–amino acid peptide at its N-terminus, and a disulfide bond that provides additional stability [34]. Interestingly, a protective regulatory role of cGSN on mitochondria-mediated apoptosis, and ultimately on mitochondrial function, was suggested to be mediated by its direct interaction with the voltage-dependent anion channel (VDAC) [38], a major pore protein complex at the mitochondrial outer membrane that facilitates the exchange of ions and small hydrophilic molecules between mitochondria and the cytosol. In fact, cGSN overexpression abrogated the respiratory chain (RC) deficiency, loss of mitochondrial membrane potential, and cytochrome *c* release exhibited by murine models of Alzheimer’s disease associated with Aβ-induced cytotoxicity [39]. Additionally, in cellular models of MD associated with RC dysfunction, we observed an upregulation of cGSN in the mitochondrial outer membrane (mGSN), where it binds VDAC to modulate its oligomeric state as a way to prevent apoptotic cell death [40,41]. This event took place in parallel to the downregulation of secreted pGSN levels, thus resulting in significantly high mGSN:pGSN protein ratios as a hallmark of OXPHOS dysfunction [41]. Our study also showed significantly decreased pGSN levels in a small cohort of MD patients [41], in agreement with previous reports suggesting that pGSN could be a useful biomarker for metabolic and oxidative stress-related clinical conditions [42], and of general health status [43]. Unfortunately, most clinical studies lacked appropriate controls to precisely determine the sensitivity and specificity of pGSN as a disease biomarker for metabolic disorders. In this work, we have determined the diagnostic performance of pGSN in MD relative to the formerly reported biomarkers, FGF-21 and GDF-15. We show that the combination of the three biomarkers improves the diagnostic capacity of each one separately, especially in those individuals under 50 years of age.

## 2. Results

### 2.1. pGSN Levels Are Significantly Decreased in MD Patients

We used ELISA assays to evaluate the plasma concentrations of pGSN relative to those of FGF-21 and GDF-15 in three study cohorts (MD patients, non-MD patients, and healthy individuals), whose main characteristics are displayed in Table 1. Individual subjects’ data are also detailed in Appendix A. We first measured pGSN concentrations in 60 adult patients diagnosed with MD (38 females, age range 18–79 years (yrs), mean age 46.8 yrs (SD ± 15.8 yrs)), relative to 41 unrelated non-MD patients (21 females, age range 18–79 yrs, mean age 48.0 yrs (SD ± 18.1 yrs)) and 56 healthy individuals (31 females, age range 18–75 yrs, mean age 50.1 yrs (SD ± 19.5 yrs)). The MD patients were initially categorized into 4 groups according to their molecular diagnosis (Figure 1A). The median plasma concentrations of pGSN (Figure 1B), FGF-21 (Figure 1C), and GDF-15 (Figure 1D) showed no statistical differences attributable to gender.

The mean plasma concentration of pGSN in MD patients was of 233 µg/mL (range 34–960 µg/mL), with a median of 143 µg/mL (interquartile range 102–336 µg/mL), a mean of 302 µg/mL (range 59–1223 µg/mL), with a median of 240 µg/mL (interquartile range 120–376 µg/mL) in the non-MD patients, and a mean of 432 µg/mL (range 213–830 µg/mL), with a median of 399 µg/mL (interquartile range 338–515 µg/mL) in the healthy cohort. Therefore, the mean pGSN concentrations were decreased by ca. 1.9- and 1.4-fold in the MD and non-MD groups respectively, when compared with the healthy cohort. In agreement, analysis of the complete cohorts showed a statistically significant decrease of pGSN levels in the MD group compared to the healthy controls (Figure 2A). However, pGSN concentrations did not clearly differentiate between the global MD and non-MD cohorts (Figure 2A). 

The mean plasma concentration of FGF-21 in MD patients was of 440 pg/mL (range 55–2908 pg/mL), of 224 pg/mL in non-MD patients (range 1–1867 pg/mL), and of 187 pg/mL in healthy controls (range 24–727 pg/mL). Therefore, the mean FGF-21 concentrations were ca. 2.4- and 1.2-fold higher for the MD and non-MD groups respectively, when compared with the healthy cohort. FGF-21 levels statistically differentiated MD patients from healthy subjects and non-MD patients, and moderately discriminated between controls and non-MD patients as a whole (Figure 2B). 

The mean plasma concentration of GDF-15 was of 1757 pg/mL in MD patients (range 329–5047 pg/mL), of 1167 pg/mL in non-MD patients (range 246–6704 pg/mL), and of 588 pg/mL in healthy controls (range 153–1563 pg/mL). Therefore, the mean GDF-15 concentrations were ca. 3.0- and 2.0-fold higher than control levels for the MD and non-MD groups, respectively. GDF-15 levels statistically differentiated MD patients from healthy subjects and non-MD patients, but it was unable to discriminate between controls and non-MD patients as a whole (Figure 2C). 

We next calculated the diagnostic ability of pGSN, FGF-21, and GDF-15 based on ROC curve analyses of all subjects (Figure 2D–E and Table 2). Comparative analysis between the whole MD and healthy cohorts showed good diagnostic discrimination values for the three biomarkers: GDF-15 (AUC = 0.87, IC95% (0.82; 0.94)), pGSN (AUC = 0.83, IC95% (0.75; 0.92)), and FGF-21 (AUC = 0.77, IC95% (0.69; 0.86)) (Figure 2D and Table 2). Subsequent tests of equality of the ROC areas showed similar discrimination ability for the plasma levels of the three biomarkers to classify between the MD and healthy cohorts (*p*-value = 0.132). In this regard, the most efficient discrimination points were as follows (cut-off values for pGSN, FGF-21, and GDF-15 are displayed in Table 3): pGSN concentrations under 252 µg/mL diagnosed MDs with 66.10% sensitivity and 98.21% specificity; for FGF-21, a threshold value of 300 pg/mL diagnosed MDs with 61.67% sensitivity and 83.93% specificity, and for GDF-15, a threshold value of 975 pg/mL diagnosed MDs with 71.93% sensitivity and 92.86% specificity.

In contrast, the ROC curve analyses between the full MD cohort vs. non-MD patients (including all individuals regardless of their age differences and clinical symptoms, Figure 2E and Table 2) showed no discriminative ability for pGSN (AUC = 0.58, IC95% (0.47; 0.70)) to distinguish between both groups, in contrast with FGF-21 (AUC = 0.77, IC95% (0.67; 0.87)) and GDF-15 (AUC = 0.76, IC95% (0.66; 0.87)) (*p*-value = 0.045 for global comparison between AUCs of pGSN, FGF-21, and GDF-15 plasma levels). In addition, FGF-21 concentrations above 184 µg/mL diagnosed MD vs. non-MD patients with 76.67% sensitivity and 73.17% specificity, and a threshold value of 1072 pg/mL for GDF-15 obtained 70.18% sensitivity and 75.61% specificity (Table 3).

### 2.2. pGSN Significantly Discriminates Young MD Patients from Healthy Controls and Non-MD Patients

Next, we categorized the individuals from the healthy, MD, and non-MD cohorts according to their age, differentiating between groups over and under 50 yrs of age (Figure 3 and Table 2; Table 3). pGSN levels significantly discriminated between MD and non-MD patients within an age range between 18 and 50 yrs (Figure 3A), as also did FGF-21 (Figure 3B) and GDF-15 (Figure 3C). Importantly, no significant differences were found between pGSN, FGF-21, and GDF-15 in their ability to distinguish the young MD patient cohort from both young controls and non-MD patients (Table 2, global *p*-values = 0.0979 and 0.2012, respectively). 

In accordance, the AUC values obtained from ROC curve analysis of the young MD and healthy cohorts (Figure 3D and Table 2) showed good diagnostic values for GDF-15 (AUC = 0.93, IC95% (0.88; 0.99)), pGSN (AUC = 0.83, IC95% (0.70; 0.94)), and FGF-21 (AUC = 0.82, IC95% (0.73; 0.93)). In individuals under 50 yrs (Table 3), pGSN concentrations under 252 µg/mL best differentiated MD patients vs. healthy controls with 73.53% sensitivity and 100% specificity; for FGF-21, a threshold value of 225 pg/mL diagnosed MDs with 65.71% sensitivity and 90.91% specificity, and for GDF-15, a threshold value of 634 pg/mL diagnosed MDs with 85.29% sensitivity and 90.91% specificity. When comparing young MD vs. young non-MD patients (Figure 3E and Table 2), the diagnostic value for pGSN (AUC = 0.67, IC95% (0.51; 0.80)) and GDF-15 (AUC = 0.85, IC95% (0.72; 0.98)) markedly improved relative to the global cohorts (Figure 2E); however, the diagnostic value of FGF-21 remained stable (AUC = 0.78, IC95% (0.63; 0.92)) regardless of the age range. In this regard, a GDF-15 threshold value of 636 pg/mL most efficiently differentiated between MD and non-MD patients under 50 yrs old, with 85.29% sensitivity and 80.95% specificity (Table 3). 

In contrast, pGSN did not correctly classify between MD and non-MD patients over 50 yrs old (Figure 3A), whereas FGF-21 (Figure 3B) and GDF-15 (Figure 3C) differentiated between these groups with a similar discrimination ability (Table 2, *p*-values after Bonferroni’s correction > 0.9). When considering the most efficient discrimination points of the three biomarkers (Table 3), an FGF-21 threshold value of 300 pg/mL best differentiated between MD and non-MD patients over 50 yrs of age, with 73.91% sensitivity and 78.95% specificity. 

### 2.3. pGSN Improves Detection of MD Patients Regardless of Their Clinical Phenotypes

We also categorized the MD patients according to their molecular diagnosis (Figure 4, left panels) and clinical phenotypes (Figure 4, right panels), and compared them relative to healthy individuals. pGSN concentrations (Figure 4A) preferentially differentiated between healthy donors and clinical cases associated with point mutations in mtDNA (including the m.3243A>G mutation), defects in mtDNA maintenance genes, and single mtDNA deletions, which were all significantly lower than the mean control value. However, pGSN levels did not differentiate between controls and MD patients with multiple mtDNA deletions. When the MD patients were grouped according to their clinical phenotypes, all groups showed significantly low pGSN levels compared to the control cohort (Figure 4B), except for those who exclusively exhibited diabetes mellitus and hearing loss.

Regarding FGF-21, it preferentially discriminated patients harboring point mutations in mtDNA, as well as mutations in mtDNA maintenance-related genes. In contrast, no significant differences in FGF-21 concentrations were found between healthy controls and patients carrying either single or multiple mtDNA deletions (Figure 4C). When categorizing the MD patients according to their clinical phenotypes (Figure 4D), we observed that FGF-21 levels were preferentially increased in patients with myopathy, exercise intolerance, and CPEO, and also discriminated patients with diabetes mellitus and hearing loss. However, FGF-21 was unable to discriminate between patients with ‘CPEO plus’ phenotypes, MELAS disease, and the 10 patients categorized as ‘other clinical phenotypes’, and healthy individuals. Notably, pGSN improved the detection of MD patients within these subgroups when compared with FGF-21 (Figure 4B), as well as that of MD patients with single mtDNA deletions (Figure 4A).

In addition, significantly elevated GDF-15 concentrations were observed in all the MD categories, regardless of their genetic diagnosis (Figure 4E) and clinical phenotypes (Figure 4F). In particular, patients with confirmed mitochondrial myopathy showed on average the highest GDF-15 levels [25], and contrary to FGF-21, GDF-15 concentration was ca. 3 times higher in patients presenting with MELAS (Figure 4F). Overall, these data suggest that GDF-15 levels most efficiently discriminate between MD patients, regardless of their clinical and genetic phenotypes, and healthy individuals.

### 2.4. Diagnostic Performance of pGSN, FGF-21, and GDF-15 Plasma Levels for Mitochondrial Disorders

Next, we assessed the statistical interdependence between pGSN, FGF-21, and GDF-15 for the MD cohort alone, as well as for the three combined cohorts, using the Spearman rank correlation coefficient (r_s_). When considering the MD cohort alone, we neither found any statistical correlation between pGSN and the previously reported biomarkers (pGSN and FGF-21: r_s_ = −0.161, *p* = 0.24; pGSN and GDF-15: r_s_ = −0.001, *p* = 0.99), nor between FGF-21 and GDF-15 (r_s_ = 0.105, *p* = 0.44). This is probably due to the wide genetic and phenotypical heterogeneity of our MD patient cohort, which impeded generating a combined ROC curve of pGSN, FGF21, and GDF-15, despite that all three biomarkers were clearly altered in most patients. In contrast, when considering the three cohorts, we observed a low negative correlation between pGSN and the previously reported biomarkers (r_s_ = −0.171 (*p* = 0.036) for pGSN and FGF-21, r_s_ = −0.247 (*p* = 0.002) for pGSN and GDF-15). In this group, FGF-21 and GDF-15 exhibited a positive moderate correlation of r_s_ = 0.340 (*p* < 0.001), similar to previous studies [5,44]. 

Finally, using a logistic regression model, we estimated that the combination of GDF-15 and pGSN achieved the best diagnostic performance to differentiate between the global cohorts of MD patients vs. healthy controls, with an AUC = 0.94, CI95% (0.89; 0.98), even better than the combined use of GDF-15 and FGF-21 (AUC = 0.91, CI95% (0.85; 0.96)). The predictive probability (*p*) of diagnosing MD patients was thus determined by the following equation:
p=1/1[1+exp(0.4827−0.0036GDF15+0.0085 pGSN)]

Predictive probabilities above 0.43 classified the MD patients with a sensitivity of 89.29%, CI95% (81.18; 97.39), and a specificity of 92.86%, CI95% (86.11; 99.60) (Table 3). According to the equation, a patient with 320 of pGSN and 1093 of GDF15 would receive a predictive probability of 0.67, and the patient would thus be classified within the MD group. To further differentiate between MD and non-MD patients, the plasma levels of either one of the two biomarkers GDF-15 or FGF-21 could be used, since the addition of both biomarkers did not significantly increase their test discrimination ability. 

## 3. Discussion

In this work, we presented pGSN as a novel biomarker that significantly reinforces the diagnostic accuracy for MD of the formerly reported biomarkers, FGF-21 and GDF-15. This conclusion is based on the following observations: (i) pGSN concentration was significantly decreased in MD patients: ROC curve analyses and logistic regression models present pGSN as a sensitive parameter capable of differentiating between MD patients and healthy individuals with a probability similar to those of FGF-21 and GDF-15. (ii) The combination of GDF-15 and pGSN achieved the best diagnostic performance to differentiate between the global cohorts of MD patients vs. healthy controls, better than either marker alone or than the combination of GDF-15 with FGF-21. (iii) In the group under 50 yrs of age, pGSN also significantly discriminated MD patients from non-MD patients with similar efficiency to FGF-21 and GDF-15. 

The average pGSN concentration in healthy subjects measured in this study (432 µg/mL) was comparable to previously documented values [43,45], and the average pGSN concentration was significantly reduced in our MD cohort. Despite its sensitivity, we observed overlapping pGSN values between MD and non-MD pathologies when all participants were considered, regardless of their age and clinical phenotypes. In agreement, decreased concentration of pGSN was previously associated with clinical deterioration in a number of unrelated pathologies associated with metabolic and oxidative stress conditions [43,46,47,48,49,50,51,52,53], as well as with ageing [54], thus behaving as a general indicator of the health status of an individual [43]. Accordingly, pGSN is the biomarker that best distinguishes the non-MD group from the healthy controls in our cohort. This ability to discriminate health status could be used to demonstrate the efficacy of different therapeutic interventions in disorders, such as MD, in which its values have been shown to decrease significantly.

It must be noted that increased levels of both FGF-21 and GDF-15 have been often associated with a variety of non-mitochondrial pathologies [17,18,19,28,29]. In fact, their utility in the diagnosis of pediatric cases of MD has been questioned, since in some cohorts they do not differentiate MD from other multisystemic inherited metabolic diseases, particularly those with liver involvement [30]. Therefore, the diagnostic accuracy of these biomarkers may vary among different studies, depending on the disease control groups used. This highlights the relevance of adding new biomarkers to the predictive model for stratification of patients with suspected MD.

In this regard, the data obtained from our three cohorts confirm the usefulness of both GDF15 and FGF21 for differentiating MD patients in adulthood, the former being the biomarker that best discriminates MD cases regardless of their genotype or clinical phenotype. Recent publications already suggested that GDF-15 outperforms FGF-21, as well as other classical MD biomarkers such as lactate, pyruvate, or CK [5,6,23,26,31]. In our study, increased plasma levels of both FGF-21 and GDF-15 significantly differentiated the MD patients from the healthy and non-MD control groups, both with statistically similar diagnostics ability after Bonferroni´s adjustment. These observations confirm the relevant diagnostics value of both biomarkers [21,23,55]. The average plasma values of FGF-21 and GDF-15 reported here also correlated with previous studies in adult MD cohorts (ranging between 7 and 8800 pg/mL for FGF-21, and 100–13,500 pg/mL for GDF-15), and these were significantly higher in the MD patients relative to the healthy and non-MD cohorts, as reported elsewhere [10,21,23,55]. The main difference between them is that the predictive value of FGF-21 seems to be restricted to patients with mitochondrial myopathies, as well as to those carrying point mutations within mtDNA and in mtDNA maintenance-related genes, as described before [10,12,55]. In contrast, the predictive value of GDF-15 was largely extended to all the MD subgroups, regardless of their clinical and genetic phenotypes. These overall data suggest that, out of the three biomarkers, GDF-15 could in principle be considered the best predictor of MD. However, one interesting observation that would refine this statement was the different behavior displayed by each of these two single biomarkers with the age of the individuals. For instance, like pGSN, the diagnostic performance of GDF-15 also improved in discriminating MD vs. non-MD patients younger than 50 yrs. This is consistent with the fact that GDF15 increases with age and has been linked to age-related disorders [51]. In contrast, FGF-21 plasma values remained stable with age, and seemed to be particularly reliable to differentiate between MD and non-MD patients over 50 yrs of age. 

Despite the three biomarkers displaying significantly altered levels in specific subsets of MD patients, we did not find statistical correlation between pGSN, FGF-21, and GDF-15 concentrations neither in our MD patient cohort, nor in any of the MD subgroups. This lack of correlation suggests that the combination of the three biomarkers in fact represents an improvement in MD diagnosis, since if they correlated perfectly, just one biomarker would be sufficient to properly distinguish between the healthy, MD, and non-MD groups. In this regard, we showed that the combined use of pGSN and GDF-15 is the best option to discriminate MD patients from healthy individuals, whereas either GDF-15 or FGF-21 alone equally discriminates between MD and non-MD patients. This diagnostic capability would improve in subjects under 50 yrs old, in whom a combination of the three biomarkers would achieve the best stratification of control, MD, and non-MD individuals. 

Therefore, we propose the combined use of pGSN, FGF-21, and GDF-15 as biomarkers to improve the overall diagnosis accuracy of adult patients with MD. Furthermore, the ability of pGSN to identify young MD patients warrants further research to confirm its potential utility in pediatric patient series, where GDF15 and FGF21 have been shown to be less relevant in differentiating multisystemic mitochondrial disorders from other metabolic diseases [30].

In summary, we presented a new biomarker that globally improves the diagnostic capacity of GDF15 and FGF-21, especially when applied to subjects under 50 yrs of age.

## 4. Materials and Methods

### 4.1. Participants

We collected blood samples from 67 adult patients who were genetically diagnosed with mitochondrial OXPHOS disease (MD) at H12O. Seven cases were later omitted due to insufficient clinical information related to their diagnosis, and thus we finally analyzed samples from 60 patients genetically diagnosed with MD [56]. As the disease control group, we collected blood samples from 41 patients diagnosed with different neuromuscular disorders of non-mitochondrial origin (non-MD) in the H12O Neuromuscular Unit. The main clinical data from the MD and non-MD patients presented in this work are described in Appendix A, respectively. As the healthy control group (Appendix A), we selected blood samples from 56 healthy adult donors who presented specific biochemical parameters within the normal range. In particular, biochemical eligibility criteria for healthy individuals included normal reference values in blood of glucose (ranging between 70 and 110 mg/dL), creatinine (0.9–1.02 mg/dL), hemoglobin (11.4–15.1 g/dL), platelets 140–450 × 103 (mm^3^), aspartate aminotransferase (5–27 U/L), alanine aminotransferase (5–34 U/L), and alkaline phosphatase (35–105 U/L). Control-to-patient samples were sex- and age-matched whenever possible. The mean ages of the healthy controls, MD, and non-MD patients were statistically comparable. 

The MD patients were initially categorized into 4 groups according to their genetic diagnosis (Figure 1A): (i) 23 patients carried mutations in nuclear genes involved in mtDNA expression and maintenance, including 10 patients with mutations in Thymidine kinase-2 (*TK2* gene), 7 patients with mutations in the DNA polymerase subunit gamma (*POLG*), 3 patients with mutations in the Twinkle mtDNA helicase (*TWNK*), 1 patient with mutations in the ribonucleoside-diphosphate reductase subunit M2B (*RRM2B*), 1 patient with mutations in optic atrophy-1 (*OPA1* gene), and 1 patient with mutations in Mitochondrial TRNA Translation Optimization 1 (*MTO1*). (ii) Twenty-five patients carried mtDNA point mutations, with the vast majority (seventeen patients) harboring the m.3243A>G mutation in the *MT-TL1* gene. (iii) Eight patients harbored hetero-plasmic single mtDNA deletions, and (iv) 4 patients had multiple mtDNA deletions without a definitive genetic diagnosis after analyzing a panel of 20 nuclear genes associated with mtDNA maintenance defects by NGS. This panel included mtDNA replication-related genes: *POLG*, *POLG2*, *TWNK*, *TFAM*, *RNASEH1*, *MGME1*, and *DNA2*, mtDNA nucleotides supply-related genes: *TK2*, *DGUOK*, *SUCLA2*, *SUCLG1*, *ABAT*, *TYMP*, *RRM2B*, *SLC25A4*, *AGK*, and *MPV17*, and mtDNA dynamics-related genes: *OPA1*, *MFN2*, and *FBXL4*. In all four patients, there was evidence of mitochondrial dysfunction on muscle biopsy (COX-negative ragged red fibers).

In addition, regardless of their genetic defects, the MD patients were categorized according to their clinical manifestations: 7 patients reported exercise intolerance without muscle weakness, 17 patients exhibited myopathy as the most prominent clinical feature, 14 cases presented chronic progressive external ophthalmoplegia (CPEO), 9 isolated CPEO and 2 with associated myopathy, 1 with associated cardiopathy, 1 with associated polyneuropathy, and 1 with exercise intolerance. Only 3 patients presented with pure MELAS, 9 patients whose main features were diabetes mellitus or migraines and hearing loss, and 10 patients were categorized as ‘other clinical phenotypes’, including 2 patients with SANDO (sensory ataxic neuropathy–dysarthria–ophthalmoparesis syndrome), 2 patients with KSS (Kearns–Sayre syndrome), and 2 patients with retinitis pigmentosa, among others. 

The non-MD patients included 8 patients with ‘exercise intolerance’ (defined by the presence of myalgias and fatigue with slightly altered CK, and muscle biopsy with no evidence of mitochondrial myopathy), 8 patients with non-mitochondrial myopathies, 8 patients with isolated hyperCKemia, 8 patients with ‘rhabdomyolysis’ (one or more episodes of acute muscle breakdown with CKs > 10,000, or myoglobinuria or secondary renal failure), 5 patients with ALS, 2 with myasthenia, and 2 with non-mitochondrial CPEO or CPEO plus.

### 4.2. Southern Blot Analysis of Single and Multiple mtDNA Deletions

Isolated DNA from skeletal muscle biopsies was analyzed by Southern Blot for the presence of either single large-scale or multiple mtDNA deletions using mtDNA-digoxigenin probes (Roche Diagnostics, Basel, Switzerland). Heteroplasmy levels and deletion size of single mtDNA deletions were estimated by densitometry (shown in Appendix A). Confirmation of negative cases was performed through long-range PCR (LR-PCR), by amplification of mtDNA using the TaKaRa LA Taq DNA Polymerase (Takara Bio Inc., Kusatsu, Japan).

### 4.3. ELISA Assays

Specific ELISA kits to GSN (Aviscera Bioscience Inc., Santa Clara, CA, USA), FGF-21 (BioVendor, Brno, Czech Republic), and GDF-15 (R&D Systems Inc., Minneapolis, MN, USA) were used to measure the concentrations of pGSN, FGF-21, and GDF-15 in plasma samples previously collected from MD patients, non-MD controls, and healthy donors, following the manufacturer’s instructions. Briefly, plasma samples were diluted between 1:8000 and 1:12,000 for pGSN, 1:2 for FGF-21, and 1:4 for GDF-15 detection. The values from each assay were extrapolated from a log-log reference curve for pGSN and FGF-21 concentrations. Results were expressed as µg/mL for pGSN, and as pg/mL for FGF-21 and GDF-15.

### 4.4. Statistical Data Analysis

Demographic, clinical characteristics, and biomarkers plasma levels of the subjects were described for the complete series with either mean (±SD) values, or relative frequencies. Data were stratified for MD patients, non-MD patients, and healthy donors, and their distributions in age were compared with ANOVA test, in sex with ×2 statistics, and plasma biomarkers with Kruskal–Wallis test. Differences between each pair of groups in plasma biomarkers were determined using the post hoc Mann–Whitney U non-parametric test. The relationship between pGSN, age, and the MD group was studied using a linear regression model. The interactions between age and the remainder of cohorts were also evaluated. The discrimination abilities of the biomarkers’ plasma levels were studied between MD patients and healthy donors, and between MD and non-MD patients. 

Receiver operating characteristic (ROC) curves for pGSN, FGF-21, and GDF-15 were estimated and graphed by varying the cut-off points used to determine which values of the clinical procedure will be considered abnormal, and then plotting the resulting true-positive rate (sensitivity) against the corresponding false-positive rate (1−specificity). Confidence intervals to ROC curves were calculated using the asymptotic normal approximations. The optimal cut-offs were determined with the Youden index. Relevant summary diagnostic parameters, namely sensitivity, specificity, positive and negative predictive values (PPV, NPV), positive and negative likelihood ratios (PLR, NLR), and efficiency (E), were also calculated. All estimations were reported with 95% confidence intervals (CI). Finally, multivariate logistic regressions were performed to classify MD patients vs. healthy controls, and between MD vs. non-MD patients. All statistical analyses were performed using GraphPad PRISM 6 (GraphPad Software, La Jolla, CA, USA) and Stata 16 software (StataCorp. 2019. Stata Statistical Software: Release 16. College Station, TX: StataCorp LLC., TX, USA).

## Figures and Tables

**Figure 1 ijms-22-06396-f001:**
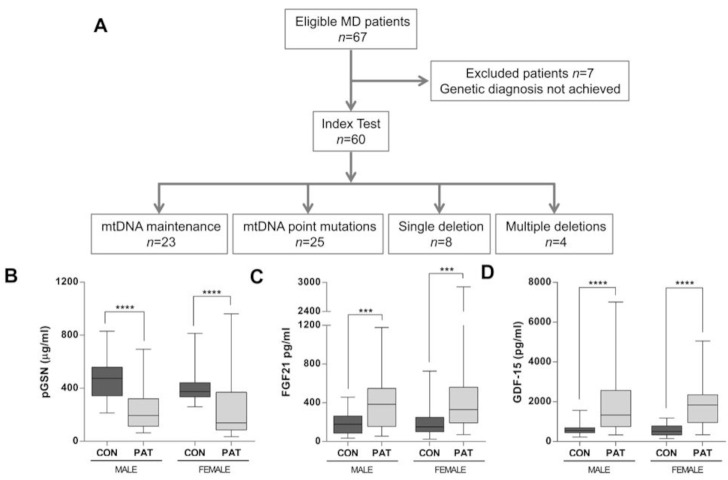
(**A**) STARD flowchart depicting the selection process of the patients with mitochondrial OXPHOS disease (MD). A total of 60 MD patients who fulfilled the genetic and clinical diagnostics criteria were enrolled in this study from 2017 to 2019. mtDNA = mitochondrial DNA. Average concentrations of (**B**) pGSN (μg/mL), (**C**) FGF-21 (pg/mL), and (**D**) GDF-15 in healthy controls (CON) vs. MD patients, stratified by sex. CON, healthy cohort; PAT, MD patient cohort. CON: Male (*n* = 24), Female (*n* = 32); MD: Male (*n* = 22), Female (*n* = 38). Data are presented as the median, interquartile range, minimum, and maximum. Mann–Whitney U test *p* values: **** *p* < 0.00001, *** *p* < 0.0001.

**Figure 2 ijms-22-06396-f002:**
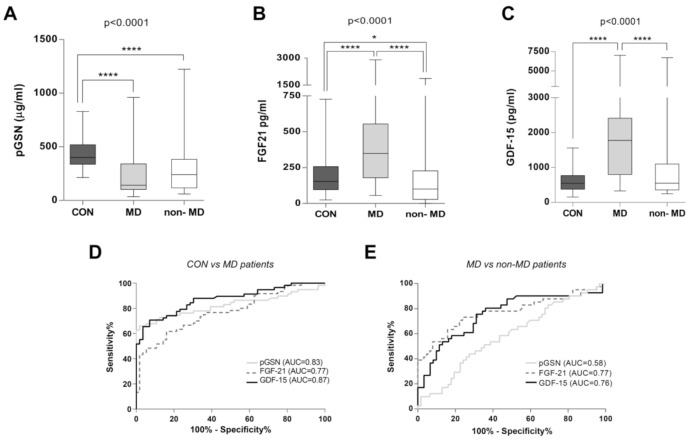
Plasma concentrations of pGSN, FGF-21, and GDF-15 in healthy controls, mitochondrial (MD), and non-mitochondrial patients (non-MD). (**A**) pGSN concentration is significantly decreased in MD and non-MD patients. (**B**) FGF-21 concentration is significantly increased in MD patients. (**C**) GDF-15 concentration is significantly increased in MD patients. For (**A**–**C**): Data represent the average biomarker concentration in healthy controls (CON, *n* = 56), MD (*n* = 60), and non-MD (*n* = 41) subjects. Data are presented as the median, interquartile range, minimum, and maximum. Kruskal–Wallis test *p*-values for the different groups are indicated in the figures. Post hoc Mann–Whitney U test *p*-values: **** *p* < 0.00001, * *p* < 0.05. For (**D**,**E**): Global ROC curves. Comparative ROC curves for pGSN, FGF-21, and GDF-15 in the full study cohorts, regardless of their age. (**D**) ROC curve in MD patients vs. healthy controls. AUC values were 0.83 for pGSN (CI: 0.75–0.91), 0.77 for FGF-21 (CI: 0.68–0.86), and 0.87 for GDF-15 (CI: 0.81–0.93). (**E**) ROC curve in MD patients vs. non-MD patients. AUC values were 0.58 for pGSN (CI: 0.47–0.70), 0.77 for FGF-21 (CI: 0.67–0.87), and 0.76 for GDF-15 (CI: 0.66–0.86). AUC = area under the curve; ROC = receiver operating characteristic; CI: confidence interval.

**Figure 3 ijms-22-06396-f003:**
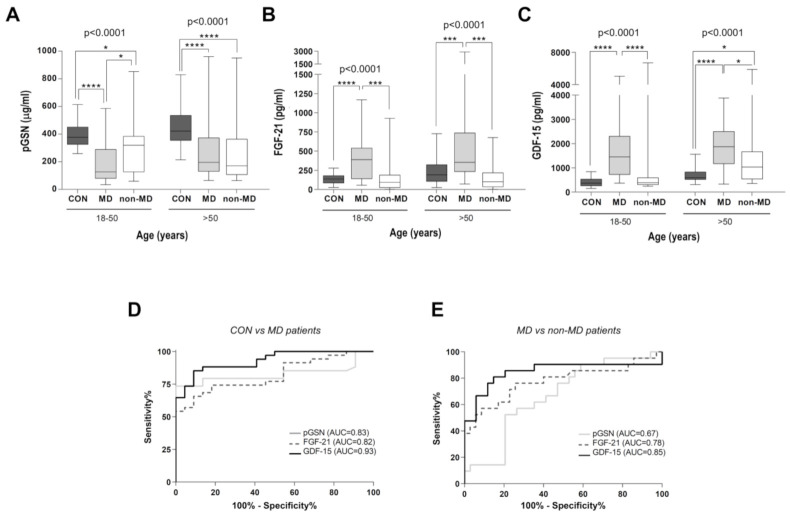
Average plasma concentrations of pGSN, FGF-2, and GDF-15 according to age. For (**A**–**C**): Average concentrations of pGSN (**B**), FGF-21 (**C**), and GDF-15 (**D**) in controls (CON, *n* = 56), MD patients (*n* = 60), and non-MD patients (*n* = 41) based on their age ranges (either between 18–50 yrs, or >50 yrs). Data are presented as the median, interquartile range, minimum, and maximum. Kruskal–Wallis test *p*-values for the two groups (subjects between 18 and 50 and >50 yrs old) are indicated on top of the figures. Post hoc Mann–Whitney U test *p*-values: **** *p* < 0.00001, *** *p* < 0.0001, * *p* < 0.05. For (**D**,**E**): Comparative ROC curves for pGSN, FGF-21, and GDF-15 in the young study cohorts (subjects between 18 and 50 yrs). (**D**) ROC curve in MD patients vs. healthy controls. AUC values were 0.83 for pGSN (CI: 0.71–0.94), 0.82 for FGF-21 (CI: 0.71–0.93), and 0.93 for GDF-15 (CI: 0.86–0.99). (**E**) ROC curve in MD patients vs. non-MD patients. AUC values were 0.67 for pGSN (CI: 0.52–0.81), 0.78 for FGF-21 (CI: 0.64–0.92), and 0.85 for GDF-15 (CI: 0.72–0.98). AUC = area under the curve; ROC = receiver operating characteristic; CI: confidence interval.

**Figure 4 ijms-22-06396-f004:**
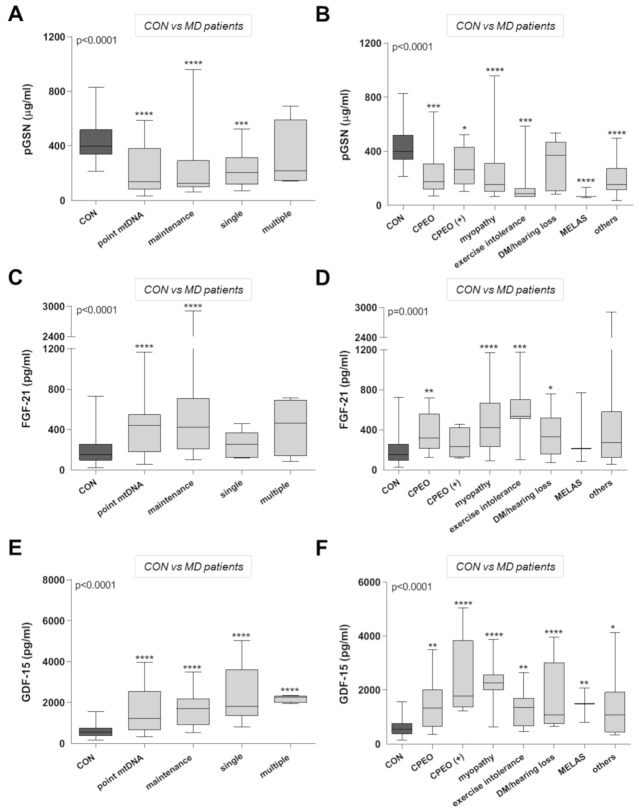
Average plasma concentrations of pGSN, FGF-21, and GDF-15 in MD patients categorized according to their genetic and clinical features. Average concentrations of pGSN (**A**), FGF-21 (**C**), and GDF-15 (**E**) in MD patients, categorized according to their genetic diagnosis. Point mtDNA = patients with mtDNA point mutations (*n* = 25), mtDNA maintenance = mutations in nuclear genes affecting mtDNA expression or maintenance (*n* = 23), single mtDNA deletion (*n* = 8), and multiple mtDNA deletions (*n* = 4). (**B**) pGSN, (**D**) FGF-21, and (**F**) GDF-15 concentration in MD patients categorized according to their clinical phenotypes. CPEO = Chronic Progressive External Ophthalmoplegia (*n* = 9); CPEO plus (*n* = 5); Myopathy (*n* = 17); Exercise Intolerance (*n* = 7); MELAS = Mitochondrial encephalopathy, lactic acidosis, and stroke-like episodes (*n* = 3); diabetes mellitus (DM) with hearing loss (*n* = 9); others = other phenotypes (*n* = 10). For (**A**–**F**): Data are presented as the median, interquartile range, minimum, and maximum. Kruskal–Wallis test *p*-values for the different groups are indicated in the figures. Post hoc Mann–Whitney U test *p*-values: **** *p* < 0.00001, *** *p* < 0.0001, ** *p* < 0.001, * *p* < 0.05.

**Table 1 ijms-22-06396-t001:** Demographics and plasma levels of pGSN, FGF-21, and GDF-15 in the sample study.

Patient		MD (*n* = 60)	Non-MD (*n* = 41)	Control (*n* = 56)	*p*-Value
**Age, mean (SD)**		46.8 (15.8)	48.0 (18.1)	50.1 (19.5)	0.61
**Gender**	F	38 (63%)	21 (51%)	31 (55%)	0.45
	M	22 (37%)	20 (49%)	25 (45%)	
**pGSN, median** **[p25, p75]**		143.0 [102.1; 335.7]	240.0 [120.4; 375.5]	399.0 [338.0; 515.3]	<0.001
FGF21, median [p25, p75]		353.1 [198.9; 588.0]	100.5 [34.2; 219.0]	154.4 [98.1; 254.9]	<0.001
**GDF15, median** **[p25, p75]**		1777.0 [800.9; 2292.0]	550.6 [361.9; 1061.6]	546.4 [376.7; 761.1]	<0.001

M: Male; F: Female; MD: mitochondrial disease; non-MD: non-mitochondrial disease; Control: healthy subjects. Median (P50), 25th percentile (P25) and 75th percentile (P75) are indicated.

**Table 2 ijms-22-06396-t002:** Discrimination ability of plasma biomarkers pGSN, FGF-21, and GDF-15.

Study Cohorts		Area Under the ROC Curve Values			^1^ Global *p*-Value	^2^*p*-Value (^3^ Adjusted *p*-Value after Bonferroni’s Correction)		
		pGSN	FGF21	GDF15		pGSN vs. FGF-21	pGSN vs. GDF-15	FGF-21 vs. GDF-15
All subjects	MD vs. CON	**0.83** (0.75;0.92)	**0.77** (0.68; 0.86)	**0.87** (0.82; 0.94)	0.1320	0.2969 (0.5939)	0.3919 (0.7838)	0.0443 (0.0886)
	MD vs. Non-MD	0.58 (0.47; 0.70)	**0.77** (0.67; 0.87)	**0.76** (0.66; 0.87)	0.0457	0.0227 (0.0455)	0.0282 (0.0565)	0.9308 (0.9999)
	Non-MD vs. CON	**0.75** (0.64; 0.86)	0.61 (0.49; 0.74)	0.57 (0.45; 0.69)	0.0553	0.0981 (0.1961)	0.0306 (0.0612)	0.6577 (0.999)
Subjects > 50 yrs old	MD vs. CON	**0.84** (0.72; 0.96)	**0.74** (0.60; 0.87)	**0.87** (0.76; 0.98)	0.1921	0.2038 (0.4076)	0.6642 (0.9999)	0.0871 (0.1742)
	MD vs. Non-MD	0.49 (0.31; 0.67)	**0.76** (0.61; 0.91)	**0.72** (0.56; 0.88)	0.0763	0.0245 (0.0489)	0.0760 (0.1521)	0.6968 (0.9999)
	Non-MD vs. CON	**0.79** (0.63; 0.94)	0.61 (0.43; 0.79)	**0.71** (0.56; 0.87)	0.3327	0.1385 (0.2770)	0.5132 (0.9999)	0.4062 (0.8123)
Subjects ≤ 50 yrs old	MD vs. CON	**0.83** (0.70; 0.94)	**0.82** (0.73; 0.93)	**0.93** (0.88; 0.99)	0.0979	0.8768 (0.9999)	0.0943 (0.1887)	0.0953 (0.9999)
	MD vs. Non-MD	**0.67** (0.51; 0.80)	**0.78** (0.63; 0.92)	**0.85** (0.72; 0.98)	0.2012	0.2764 (0.5529)	0.0738 (0.1476)	0.4764 (0.9528)
	Non-MD vs. CON	**0.69** (0.53; 0.86)	0.61 (0.43; 0.79)	0.56 (0.39; 0.74)	0.4969	0.4289 (0.8579)	0.2805 (0.5610)	0.7348 (0.9999)

^1^ Global *p*-values of the test of equality of the three ROC areas obtained from applying pGSN, FGF21, and GDF15 to independent cohorts. All classifiers have equal AUC values. ^2^
*p*-value and ^3^ adjusted *p*-value after Bonferroni’s correction for multiple comparisons obtained from the tests of equality of ROC areas of the indicated biomarkers. Relevant diagnostic AUC values are indicated in bold. Grey background highlights statistically significant differences between biomarkers.

**Table 3 ijms-22-06396-t003:** General features of the plasma biomarkers pGSN, FGF-21, and GDF-15.

Study CohortsS	Cut-Off Values	Sensitivity% (95% CI)	Specificity% (95% CI)	PPV% (95% CI)	NPV% (95% CI)	PLR (95% CI)	NLR (95% CI)	Efficiency %
All subjects	MD vs. Control							
	pGSN ≤ 252	66.10 (54.02; 78.18)	**98.21** (94.75; 100)	**97.50** (92.66; 100)	73.33 (63.33; 83.34)	36.92 (5.24; 259.79)	0.34 (0.24; 0.49)	**81.74**
	FGF-21 ≥ 300	61.67 (49.36; 73.97)	**83.93** (74.31; 93.55)	**80.43** (68.97; 91.90)	67.14 (56.14; 78.15)	3.83 (2.04; 7.21)	0.45 (0.32; 0.64)	72.41
	GDF-15 ≥ 975	71.93 (60.26; 83.59)	**92.86** (86.11; 99.60)	**91.11** (82.80; 99.43)	76.47 (66.39; 86.55)	10.07 (3.86; 26.26)	0.30 (0.19; 0.46)	**82.30**
	**Model (pGSN + GDF-15) ≥ 0.43**	**89.29** (81.18; 97.39)	**92.86** (86.11; 99.60)	**92.59** (85.61; 99.58)	**89.66** (81.82; 97.49)	12.50 (4.84; 32.29)	0.11 (0.05; 0.24)	**91.07**
	MD vs. non-MD							
	FGF-21 ≥ 184	76.67 (65.96; 87.37)	73.17 (59.61; 86.73)	**80.70** (70.46; 90.95)	68.18 (54.42; 81.94)	2.85 (1.69; 4.82)	0.31 (0.19; 0.52)	75.25
	GDF-15 ≥ 1072	70.18 (58.30; 82.05)	75.61 (62.46; 88.75)	**80.00** (68.91; 91.09)	64.58 (51.05; 78.11)	2.87 (1.63; 5.06)	0.39 (0.25; 0.60)	72.45
Subjects > 50 yrs old	MD vs. Control							
	pGSN ≤ 320	69.57 (50.76; 88.37)	**88.24** (77.41; 99.07)	**80.00** (62.47; 97.53)	**81.08** (68.46; 93.70)	5.91 (2.26; 15.44)	0.34 (0.18; 0.64)	**80.70**
	FGF-21 ≥ 300	73.91 (55.97; 91.86)	73.53 (58.70; 88.36)	65.38 (47.10; 83.67)	**80.65** (66.74; 94.55)	2.79 (1.51; 5.14)	0.35 (0.17; 0.72)	73.68
	GDF-15 ≥ 1227	**80.95** (64.16; 97.75)	**94.12** (86.21; 100)	**89.47** (75.67; 100)	**88.89** (78.62; 99.15)	13.76 (3.53; 53.66)	0.20 (0.08; 0.49)	**89.09**
	MD vs. Non-MD							
	FGF-21 ≥ 300	73.91 (55.97; 91.86)	78.95 (60.62; 97.28)	**80.95** (64.16; 97.75)	71.43 (52.11; 90.75)	3.51 (1.42; 8.67)	0.33 (0.15; 0.68)	76.19
	GDF-15 ≥ 1707	**80.95** (64.16; 97.75)	47.37 (24.92; 69.82)	62.96 (44.75; 81.18)	69.23 (44.14; 94.32)	1.53 (0.95; 2.47)	0.40 (0.14; 1.09)	65.00
	Non-MD vs. Control							
	pGSN ≤ 240	63.16 (41.47; 84.45)	**97.06** (91.38; 100)	**92.31** (77.82; 100)	**82.50** (70.72; 94.28)	21.48 (3.02; 152.70)	0.37 (0.21; 0.68)	**84.91**
Subjects ≤ 50 yrs old	MD vs. Control							
	pGSN ≤ 252	73.53 (58.70; 88.36)	**100** (100; 100)	**100** (100; 100)	70.97 (54.99; 86.95)	na	0.26 (0.15; 0.46)	**83.93**
	FGF-21 ≥ 225	65.71 (49.99; 81.44)	**90.91** (78.90; 100)	**92.00** (81.37; 100)	62.50 (45.73; 79.27)	7.22 (1.88; 27.68)	0.37 (0.23; 0.60)	75.44
	GDF-15 ≥ 634	**85.29** (73.39; 97.20)	**90.91** (78.90; 100)	**93.55** (84.90; 100)	**80.00** (64.32; 95.68)	9.38 (2.48; 35.43)	0.16 (0.07; 0.36)	**87.50**
	MD vs. Non-MD							
	pGSN ≤ 310	79.41 (65.82; 93.00)	52.38 (31.02; 73.74)	72.97 (58.66; 87.28)	61.11 (38.59; 83.63)	1.66 (1.03; 2.69)	0.39 (0.18; 0.85)	69.09
	FGF-21 ≥ 166	74.29 (59.81; 88.77)	76.19 (57.97; 94.41)	**83.87** (70.92; 96.82)	64.00 (45.18; 82.82)	3.12 (1.41; 6.87)	0.33 (0.18; 0.62)	75.00
	GDF-15 ≥ 636	**85.29** (73.39; 97.20)	**80.95** (64.16; 97.75)	**87.88** (76.74; 99.01)	77.27 (59.76; 94.78)	4.47 (1.83; 10.93)	0.18 (0.07; 0.41)	**83.64**
	Non-MD vs. Control							
	pGSN ≤ 245	73.53 (58.70; 88.36)	**100** (100; 100)	**100** (100; 100)	70.97 (54.99; 86.95)	na	0.26 (0.15; 0.46)	**83.93**

MD: mitochondrial disease; non-MD: non-mitochondrial disease; Control: healthy subjects; CI: confidence interval; PPV: positive predictive value; NPV: negative predictive value; PLR: positive likelihood ratio; NLR: negative likelihood ratio. The following cut-off points were not included due to their lack of discriminative performance between groups: of pGSN, FGF-21, and GDF-15 for non-MD patients vs. Control (all subjects); of pGSN for MD vs. non-MD patients (subjects > 50 yrs old); of FGF-21 and GDF-15 for non-MD patients vs. Control (subjects > 50 yrs old); of FGF-21 and GDF-15 for non-MD patients vs. Control (subjects ≤ 50 yrs old). Predictive values above 80% are indicated in bold. na, not applicable.

## Data Availability

Correspondence and material requests should be addressed to the corresponding author, C.U.

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
