# Peer review of "Plasma Gelsolin Reinforces the Diagnostic Value of FGF-21 and GDF-15 for Mitochondrial Disorders"

_ijms, 2021, doi:10.3390/ijms22126396_

Round 1

Reviewer 1 Report

This study of Cristina Ugalde's Lab represents very important step in searching for perspective biomarkers that would improve noninvasive diagnostics and screening of human mitochondrial diseases. Their previous studies showed that plasma isoform of cytoskeletal protein Gelsolin (pGSN) specfically decreases in cellular models of mitochondrial OXPHOS deficiency in conjunction with anti-apoptotic response and accumulation of cellular Gelsolin in mitochondria. In the present extensive clinical study they investigated broad range of different types of mitochondrial disorders caused by mtDNA or nuclear gene mutations affecting respiratory chain complexes I, III, IV or ATP synthase versus healthy donors and non-mitochondrial neuromuscular cases.  They tested diagnostic potential of pGSN in compartison with established cytokine biomarkers FGF-21 and GDF-15 and clearly showed that pGSN can improve accuracy of diagnostics of mitochondrial disorders, especially in younger subjects under 50 years of age. The study suggests to use combinantion of pGSN, GDF-15 and FGF-21 to further improve diagnostics of mitochondrial disorders in adult patients.  

Author Response

We thank the reviewer very much for his/her positive assessment of our work.

Reviewer 2 Report

The manuscript by Penas and colleagues investigated plasma Gelsolin (pGSN) as a diagnostic marker for Mitochondrial disorder (MD). Authors suggested that combination of pGSN, GDF-15, and FGF-21 could provide highest diagnostic accuracy in stratification of control, MD and 403 non-MD individuals. The strength of this study is that authors used significant amount of human samples. The study is interesting and provided deep insight for the diagnosis of MD.

Following queries should be resolved for the improvement of this manuscript.

  1. In method and materials, section 4.1 participants, and also In Table S1, authors did not disclose the mutation location of some genes like TK2,  PLOG, TWNK, RRM2B, OPA1 Authors should disclose the locations of the mutations of all genes, authors mentioned in this manuscript. Please also disclose the location of single and multiple deletion of mtDNA mentioned in the table S1.
  2. It would of interest to discuss about the Gelsolin in relation to the mitochondrial function. Authors should describe little about molecular perspective of Gelsolin and mitochondrial function.

Overall, the manuscript is well written, organized and interesting.

Author Response

We thank the reviewer for his/her positive assessment of our work. We believe that the revised manuscript addresses all of the comments raised by the Reviewer and it has been modified according to the proposed recommendations.

Point-by-point response to the reviewer´s queries:

1) In method and materials, section 4.1 participants, and also In Table S1, authors did not disclose the mutation location of some genes like TK2, PLOG, TWNK, RRM2B, OPA1. Authors should disclose the locations of the mutations of all genes, authors mentioned in this manuscript. Please also disclose the location of single and multiple deletion of mtDNA mentioned in the table S1.

Response: As indicated by the Reviewer, we have disclosed in Table S1 the mutation location of all nuclear genes affected in our patients. However, we could not disclose the precise location of the breakpoints in mtDNA leading to single and multiple mtDNA deletions because they were exclusively confirmed for diagnostic purposes by Southern-blot analyses, without further mapping by sequencing of the specific breakpoints. The Southern-blot method used for detection of mtDNA deletions has now been described in the Materials and Methods Section. Since all single mtDNA deletions were found in heteroplasmy, their heteroplasmic levels (in %) as well as the mtDNA deletion size have also been disclosed in Table S1. In addition, for consistency of data presentation, we have added to Table S1 the heteroplasmic levels of point mutations in muscle mtDNA when available. It was not possible to map the specific breakpoints of the multiple mtDNA deletions due to their multivariate nature.

2) It would of interest to discuss about the Gelsolin in relation to the mitochondrial function. Authors should describe little about molecular perspective of Gelsolin and mitochondrial function.

Response: According to the Reviewer´s recommendation, we have extended in the Introduction the putative role of Gelsolin on mitochondrial function.